# Accurate statistical methods to cover the aspects of the increase in the incidence of kidney failure: A survey study in Ha'il -Saudi Arabia

**Alanazi Talal Abdulrahman** [ORCID][1]*, **Dalia Kamal Alnagar**[2]

**1** Department of Mathematics, University of Ha'il, Ha'il, Saudi Arabia, **2** Statistics Department, University of Tabuk, Tabuk, Saudi Arabia

* t.shyman@uoh.edu.sa

## Abstract

### Introduction

Chronic kidney disease (CKD) has become more common in recent decades, putting significant strain on healthcare systems worldwide. CKD is a global health issue that can lead to severe complications such as kidney failure and death.

### Objective

The purpose of this study was to investigate the actual causes of the alarming increase of kidney failure cases in Saudi Arabia using the supersaturated design analysis and edge design analysis.

### Materials and methods

A cross-sectional questionnaire was distributed to the general population in the KSA, and data were collected using Google Forms. A total of 401 responses were received. To determine the actual causes of kidney failure, edge and supersaturated designs analysis methods were used, which resulted in statistical significance. All variables were studied from factor $h_1$ to factor $h_{18}$ related to the causes of kidney failure.

### Results

The supersaturated analysis method revealed that the reasons for the increase in kidney failure cases are as follows: $h_9$(Bad diet), $h_8$(Recurrent urinary tract infection), $h_1$ (Not drinking fluids), $h_6$ (Lack of exercise), $h_{14}$ (drinking from places not designated for valleys and reefs), $h_{18}$ (Rheumatic diseases), $h_{10}$ (Smoking and alcohol consumption), $h_{13}$ (Direct damage to the kidneys), $h_2$ (take medications), $h_{17}$ (excessive intake of soft drinks), $h_{12}$ (Infection), $h_5$ (heart disease), $h_3$ (diabetes), $h_4$ (pressure disease), $h_{15}$ (Dyes used in X-rays), and $h_{11}$ (The presence of kidney stones) are all valid. The design analysis method by edges revealed that the following factors contributed to an increase in kidney failure cases: $h_8$ (Recurrent urinary tract infection), $h_6$ (Lack of exercise), $h_7$ (Obesity), and $h_{11}$.

**Data Availability Statement:** The authors confirm that the data supporting the findings of this study are available within the article [and/or] its supplementary materials.

**Funding:** The authors received no specific funding for this work.

**Competing interests:** The author states that I have no conflicts of interest.

## Conclusion

The findings showed that there were causes of kidney failure that led to the statistical significance, which is $h_8$ (Recurrent urinary tract infection) and $h_{11}$ (The presence of kidney stones)

## 1. Introduction

The kidney is considered one of the most essential parts of the human body as it acts as a filter to purify fluids and blood from impurities eliminate waste and toxic substances in the blood and excrete them outside the body through urine, in addition to controlling the number of fluids, sodium and potassium present in the body. Kidney failure occurs when the kidneys cannot effectively eliminate the waste products. Kidneys may lose the ability to filter waste and excrete liquid waste through urine, resulting in a chronic or acute condition known as kidney failure. In addition, it causes an imbalance in the level of water, mineral salts, and various minerals in the body, which leads to disturbances in the body's systems, it may threaten life if it is not treated immediately [1]. Chronic kidney disease (CKD) has become more common in recent decades, putting a significant strain on healthcare systems worldwide. CKD is a global health issue that can lead to severe complications such as kidney failure and death. It affects 195 million women worldwide annually and is currently the eighth leading cause of death in women, accounting for 600,000 deaths each year [2]. Patients with end-stage kidney disease (ESKD) have a 17-fold higher mortality rate than age- and sex-matched healthy people. 5 The number of deaths from CKD is expected to reach 2–4 million by 2040 [3]. According to an epidemiological survey conducted in 2010, the global prevalence of CKD was 9.1%, with 697.5 million cases of CKD (all stages) reported worldwide. In contrast, the prevalence of CKD in the Kingdom of Saudi Arabia is 5.7%, posing a significant burden on the healthcare systems. In recent years, the medical literature and community have widely accepted that CKD is associated with an increased risk of premature death [4]. The Executive Director General of the Prince Salman Center for Kidney Diseases and the General Supervisor of the Awareness Campaign for Kidney Diseases, Dr. Khaled bin Abdulaziz Al-Saaran, revealed that the incidence of kidney failure in the Kingdom ranges from 90 to 110 people per million people in the Kingdom who suffer from kidney failure. The incidence of kidney failure in the northern part of the kingdom is the highest compared that on to other regions of the Kingdom, reaching 167 per million people. Some studies indicate that the global incidence of kidney disease is one out of every ten healthy people. The latest kidney failure statistics showed that the total number of patients with chronic renal failure reached 21,000 in Saudi Arabia. According to the Saudi Center for Organ Transplantation annual report, most patients were men 56% and women 44%. During our simple survey ten years ago, the number of people with kidney failure in Saudi Arabia was approximately 9,600 in Saudi [5].

Compared with latest statistics, we find that the number is increasing significantly and is being observed by the competent authorities. However, we did not find a survey study that looked for the reasons for the increase in cases of kidney failure worldwide, as researchers were limited during the past years to treat only in the advanced stages and urging early detection of this disease.

This study aimed to identify the reasons for the increase in kidney failure cases by conducting a survey using various statistical models. Additionally, this study examines two methods, a

supersaturated design and an edge design supersaturated design, to identify the actual reasons for the increase in kidney failure cases.

Supersaturated Design Analysis is a statistical approach used in experiments in which the number of factors exceeds the number of runs. This is useful when it is believed that only a few factors are significant and particularly beneficial for screening purposes. These designs are known for their run-size economy and have been proven to be effective in identifying significant factors [6–8]. Edge design analysis refers to the study and evaluation of experimental designs that are particularly useful for screening experiments with more factors than runs. These designs help identify the most influential factors with a limited number of experiments; in addition, the analysis of edge designs often involves statistical methods to assess the robustness and efficiency of the designs [9].

The validity of the two methods, Analysis by supersaturated designs and analysis by design with edges, can be assessed based on their effectiveness in identifying significant factors in an experimental setup.

## 2. Materials and methods

### 2.1. Survey study

This section contains general questions related to metadata and causes of kidney failure. The general questions included gender, age, region, a chronic disease, and kidney failure. Regarding the questions related to the causes of kidney failure, where the opinion of the competent people, the patient, or those around the patient is taken about the actual cause from his simple point of view for the following reasons:

- $h_1$: Not drinking fluids

- $h_2$: Take medications

- $h_3$: Diabetes

- $h_4$: Pressure disease

- $h_5$: heart disease

- $h_6$: Lack of exercise

- $h_7$: Obesity

- $h_8$: Recurrent urinary tract infection

- $h_9$: Bad diet

- $h_{10}$: Smoking and alcohol consumption

- $h_{11}$: The presence of kidney stones

- $h_{12}$: Infection

- $h_{13}$ Direct damage to the kidneys

- $h_{14}$: Drinking from places not designated for valleys and reefs

- $h_{15}$: Dyes used in x-rays

- $h_{16}$: Stress and lack of sleep

- $h_{17}$: Excessive intake of soft drinks

- $h_{18}$: Rheumatic diseases

- Y: How many cases do you know of that have kidney failure?

## 2.2. The recruitment period and ethics statement

After obtaining approval from the Research Ethics Committee, the questionnaire was distributed to the target group from 01/24/2023 to 06/24/2023. The Research Ethics Committee (RCE) at the University of Ha'il reviewed and approved this study on January (23, 2023, research number H-2023-040). Verbal and written consent were obtained from all participants prior to data collection.

## 2.3. Methods used in the analysis

**2.3.1. Analysis by supersaturated designs.** Contrast method analysis with supersaturated designs was used to determine the causes of kidney failure, which were statistically significant. The procedure is as follows, [10].

1. Discover all factors that are distinct through the equation:

2. Begin with I = 0 and work your way up to p = N/2, where N is the number of trials.

3. Look for the following equations:

$$M = X^T Y, \tag{1}$$

Y is the response factor, and X is the design chosen from the graphic survey. At this point, the superior attributes and ranking factors express contracts.

$$uml_i = |m_k| - t_{k-1,\frac{z}{2}}\sigma_p \tag{2}$$

$$lml_i = -|m_k| + t_{k-1,\frac{z}{2}}\sigma_p, \tag{3}$$

where $t_{k-1,\frac{z}{2}}$ is the relative distribution of the t table.

1. Remove the highest valued $|m_{k-1}|$ and then set I = I + 1.

2. Find the $\sigma_p$ for the p most significant supreme differences using only the leftover qualities.

3. From Eqs (2) and (3), if the fluctuation in E is less than the difference found before Step 3, proceed to Step 5; otherwise, stop and close the dynamic components from the differences outside the primary district.

4. More details on this method can be found in reference [11].

**2.3.2. Analysis by design with edges.** Edge designs analysis was used to determine the actual causes of kidney failure, which resulted in the statistical significance. The procedure described in [12] is as follows.

1. Find $z_{i,j} = y_i - y_j$, (i, j) $\epsilon$ E.

2. After determined the values of the active factors, the absolute value of all values was found and arranged in descending order.

3. We start with the value p zero and find the median for all the values of the first step, considering that the values of p depend on the values of Z.

4. Find

5. $\sigma = \left| \frac{med\{z_{i,j}:(i,j)\epsilon E\}}{0.675\sqrt{2}} \right|$

6. We calculate this equation: k×2^0.5 $\sigma(p)$

Based on the previous step, we searched for the number of active w (p) agents based on the value of Z.

More details on this method can be found in reference [13].

## 2.4. Combine the results of the two methods

In this section, models and applications for each method are selected, and the analysis was used by a saturated design for each particular model of this method to search for the reasons that led to an increase in cases of kidney failure. Then, the edge analysis method was used for the aforementioned selected model and design to search for the reasons. In the end, similar causes were identified in both methods, therefore, these are the actual reasons that led to increased cases of kidney failure.

# 3. Results

## 3.1. The results of the general data analysis

This section presents the results of the questionnaire answers to the general questions related to our research.

Fig 1 shows the responses to the questionnaires according to age. The age group from 18 to 25 constituted 52 percent as the highest response rate, followed by the age group from 26 to 35 (20%), and the age group from 36 to 50 (19%). While the age group over 50 years achieved a low percentage of the questionnaire responses, at 9 percent. Fig 2 shows that the response rate of the questionnaire for males was equal to the response rate for females. Fig 3 shows the response rate for each region, as the northern region occupied the highest response rate, estimated at 52%, followed by the central area at 23%, while the rest of the regions are as shown in the figure. Fig 4 shows whether those who answered the questionnaire had chronic disease and kidney failure, where the highest percentage was that they did not have these diseases.

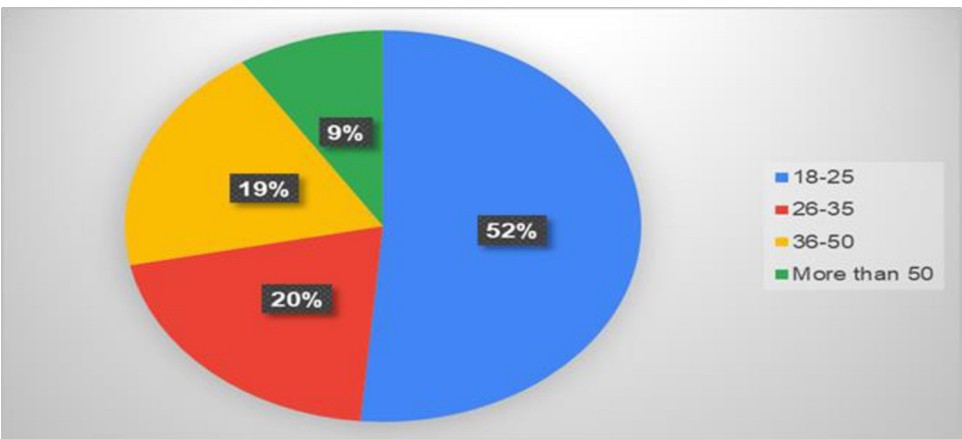

**Fig 1. The percentage of response by age.**

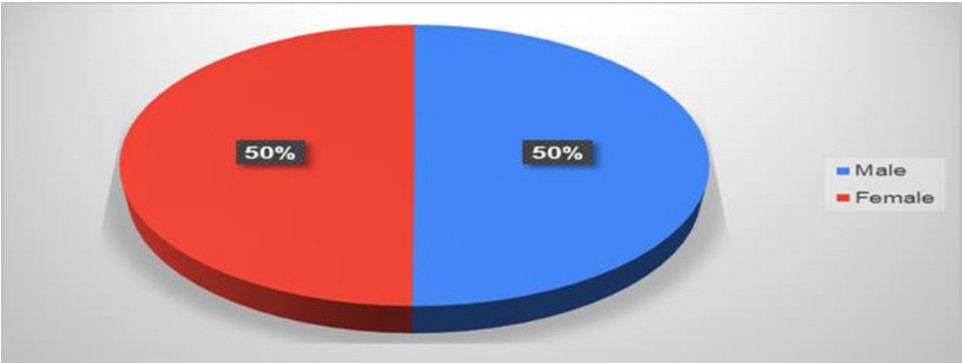

**Fig 2. The rate of gender responses.**

### 3.2. Analysis results of the supersaturated designs method

In this section, applications are made for the questionnaire, which consists of a supersaturated design so that the number of influencing factors is greater than the number of responses at one rate. The above analysis method was then used.

**3.2.1. Application 1.** We assumed that n = 17, as shown in Table 1. The authors employed the aforementioned analysis techniques. The Eq 1 used to find all factor contrasts, calculate absolutes, and sort these values. The outcomes are presented in Table 2. Set I to 0 and compute the variance of the p most significant absolute contracts (using p = N/2, where N is the run size). Table 3 shows the outcome. The authors noticed that $\sigma_2^2$ is greater than $\sigma_1^2$, so we stopped and looked for the active factors. The final values of Stage 1's are $uml_i$ = 25.10, $lml_i$ = -25.10 and $\sigma_1^2$ = 22. Consequently, the following active factors exist outside the critical region: $h_9, h_8, h_1, h_6, h_{14,}$ and $h_{18}$.

**3.2.2. Application 2.** We assume that n = 17, as listed in Table 4. The authors employed the aforementioned analysis techniques. The Eq 1 was used to find all factor contrasts, calculate absolutes, and sort these values. Table 5 shows the result. We set I to 0 and compute the variance of the p most significant absolute contracts (using p = N/2, where N is the run size). The outcomes were presented in Table 6. The authors noticed that $\sigma_2^2$ is greater than $\sigma_1^2$, so we

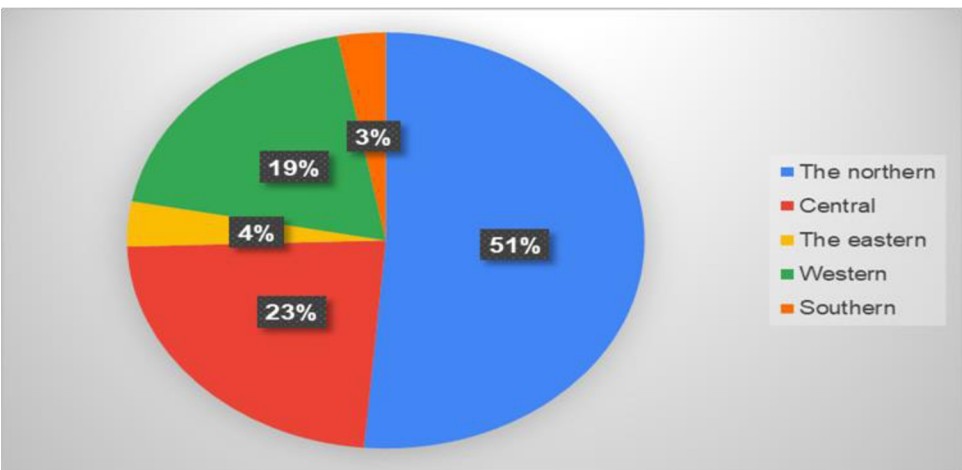

**Fig 3. The rate of region responses.**

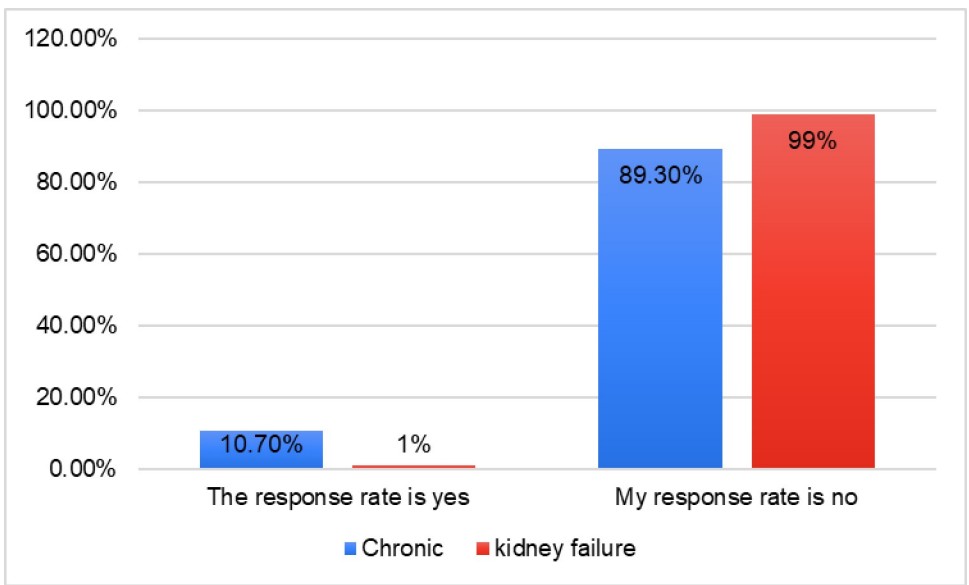

**Fig 4. Percentage of chronic disease and Kidney failure or not.**

stopped and looked for the active factors. The final values in Stage 1's is $uml_i$ = 14.35, $lml_i$ = -14.35 and $\sigma_1^2$ = 13.11. As a result, the following active factors exist outside the critical region: $h_{10}, h_{13}, h_1, h_2, h_8, h_9,$ and $h_{17}$.

**3.2.3. Application 3.** We assume that n = 17, as listed in Table 7. The authors employed the techniques mentioned above to analyze the situation. Eq 1 was used to find all factor contrasts, calculate absolutes, and sort these values. The outcomes were presented in Table 8. We set I to 0 and compute the variance of the p most significant absolute contracts (using p = N/2,

**Table 1. The first supersaturated application of the questionnaire.**

| $h_1$ | + | + | + | + | + | + | + | - | + | + | + | + | + | + | - | + | + |
|---|---|---|---|---|---|---|---|---|---|---|---|---|---|---|---|---|---|
| $h_2$ | - | + | - | + | + | - | - | - | + | + | + | + | - | - | + | + | - |
| $h_3$ | + | - | - | - | + | - | - | - | - | + | + | + | + | - | + | + | - |
| $h_4$ | + | - | - | - | + | - | - | - | - | + | + | + | + | - | - | + | + |
| $h_5$ | + | - | - | - | + | - | - | - | - | + | + | + | + | - | - | + | - |
| $h_6$ | - | + | + | - | + | - | + | - | + | + | + | + | - | + | + | + | + |
| $h_7$ | + | - | + | - | + | - | + | - | - | + | - | + | - | + | - | - | - |
| $h_8$ | + | - | + | - | + | - | + | + | + | + | - | + | + | + | + | - | + |
| $h_9$ | + | + | + | - | + | + | + | + | + | + | + | + | + | - | + | - | - |
| $h_{10}$ | + | + | + | + | + | - | + | - | + | + | + | + | + | + | - | + | - |
| $h_{11}$ | + | + | + | + | + | + | - | + | + | + | + | + | - | + | - | + | - |
| $h_{12}$ | + | - | + | + | - | - | + | - | - | + | + | + | - | - | + | + | - |
| $h_{13}$ | + | + | + | + | + | + | - | + | + | + | + | + | - | + | - | + | - |
| $h_{14}$ | + | + | + | - | + | - | - | - | + | + | + | + | + | - | + | - | - |
| $h_{15}$ | - | - | + | + | + | - | - | - | + | + | + | + | + | - | - | + | - |
| $h_{16}$ | - | - | + | + | - | + | + | - | - | - | + | + | - | - | + | - | - |
| $h_{17}$ | + | + | + | - | + | + | + | - | + | + | + | + | + | + | - | - | + |
| $h_{18}$ | - | - | - | - | - | - | - | - | - | - | + | + | - | - | + | - | - |
| Y | 2 | 3 | 8 | 4 | 7 | 2 | 2 | 1 | 2 | 5 | 1 | 3 | 2 | 1 | 10 | 0 | 2 |

**Table 2. Sorted absolute contrasts for Application 1.**

| J | 1 | 2 | 3 | 4 | 5 | 6 | 7 | 8 | 9 | 10 | 11 | 12 | 13 | 14 | 15 | 16 | 17 | 18 |
|---|---|---|---|---|---|---|---|---|---|----|----|----|----|----|----|----|----|----|
| $h_n$ | $h_9$ | $h_8$ | $h_1$ | $h_6$ | $h_{14}$ | $h_{18}$ | $h_{10}$ | $h_{17}$ | $h_{11}$ | $h_{13}$ | $h_2$ | $h_5$ | $h_{12}$ | $h_4$ | $h_{15}$ | $h_3$ | $h_{16}$ | $h_7$ |
| $|M(j)|$ | 41 | 35 | 33 | 33 | 31 | 27 | 25 | 25 | 23 | 23 | 15 | 15 | 15 | 11 | 9 | 5 | 5 | 1 |

**Table 3. Step-by-step calculations for Application 1.**

| i | $\sigma_i^2$ | $uml_i$ | $lml_i$ | $\sigma_i^2 > \sigma_{i-1}^2$ |
|---|---|---|---|---|
| 0 | 34 | 28.74 | -28.74 | - |
| 1 | 22 | 25.10 | -25.10 | No |
| 2 | 33.11 | 20.80 | -20.80 | Yes |

where N is the run size). Table 9 showed the result. The authors noticed that $\sigma_1^2$ is greater than $\sigma_0^2$, we stopped and looked for the active factors. Stage 1's final values are $uml_i = 35.19$, $lml_i = -35.19$ and $\sigma_0^2 = 21.77$. Consequently, the following active factors exist outside the critical region: $h_1, h_{17}, h_9, h_{11}, h_{13},$ and $h_8$.

**3.2.4. Application 4.** We assume that n = 17, as listed in Table 10. The authors employed the aforementioned techniques for the analysis. Eq 1 was used to find all factor contrasts,

**Table 4. The second supersaturated application of the questionnaire.**

| | | | | | | | | | | | | | | | | | | |
|---|---|---|---|---|---|---|---|---|---|---|---|---|---|---|---|---|---|---|
| $h_1$ | + | + | + | - | + | + | + | - | + | + | + | + | + | + | + | + | + | - |
| $h_2$ | + | + | + | - | + | + | + | + | + | + | + | + | + | + | + | + | + | - |
| $h_3$ | + | + | + | - | + | - | + | - | - | + | - | - | + | + | + | + | + | - |
| $h_4$ | + | + | + | - | - | - | + | - | - | + | - | - | - | + | + | + | + | - |
| $h_5$ | + | + | + | - | + | - | - | - | - | - | - | + | - | + | + | + | + | - |
| $h_6$ | + | + | + | + | - | - | - | + | - | + | - | + | + | + | + | + | + | + |
| $h_7$ | + | + | + | + | + | + | - | + | + | - | - | - | + | + | + | + | + | + |
| $h_8$ | + | + | + | - | + | + | + | + | - | - | - | + | - | + | + | + | + | - |
| $h_9$ | + | + | + | + | - | + | + | + | + | + | + | - | + | + | + | + | + | + |
| $h_{10}$ | + | + | + | - | + | + | + | + | + | - | + | + | + | + | + | + | + | - |
| $h_{11}$ | + | + | - | - | + | + | + | + | + | + | + | + | + | + | + | - | - | - |
| $h_{12}$ | + | + | - | - | - | - | - | + | - | + | - | - | - | + | + | - | - | - |
| $h_{13}$ | + | + | + | - | + | + | + | + | + | + | + | - | + | + | + | + | + | - |
| $h_{14}$ | + | + | - | - | + | + | - | + | - | + | + | + | - | + | + | + | - | - |
| $h_{15}$ | + | + | - | - | - | - | + | + | - | - | + | + | + | + | + | - | - | - |
| $h_{16}$ | + | + | - | - | - | - | - | + | - | - | - | - | + | + | + | - | - | - |
| $h_{17}$ | + | + | + | - | + | + | - | + | + | + | + | + | + | + | + | + | + | - |
| $h_{18}$ | + | + | - | - | + | - | + | - | - | + | - | - | + | + | - | - | | |
| Y | 2 | 2 | 5 | 1 | 2 | 3 | 2 | 2 | 0 | 2 | 1 | 2 | 1 | 0 | 0 | 1 | 2 | |

**Table 5. Sorted absolute contrasts for Application 2.**

| J | 1 | 2 | 3 | 4 | 5 | 6 | 7 | 8 | 9 | 10 | 11 | 12 | 13 | 14 | 15 | 16 | 17 | 18 |
|---|---|---|---|---|---|---|---|---|---|----|----|----|----|----|----|----|----|----|
| $h_n$ | $h_{10}$ | $h_{13}$ | $h_1$ | $h_2$ | $h_8$ | $h_9$ | $h_{17}$ | $h_7$ | $h_{12}$ | $h_{14}$ | $h_5$ | $h_{11}$ | $h_3$ | $h_{18}$ | $h_6$ | $h_4$ | $h_{15}$ | $h_{16}$ |
| $|M(j)|$ | 24 | 22 | 20 | 20 | 18 | 18 | 18 | 14 | 12 | 12 | 10 | 10 | 8 | 8 | 6 | 4 | 4 | 4 |

**Table 6. Step-by-step calculations for Application 2.**

| i | $\sigma_i^2$ | $uml_i$ | $lml_i$ | $\sigma_i^2 > \sigma_{i-1}^2$ |
|---|---|---|---|---|
| 0 | 13.77 | 16.20 | -16.20 | - |
| 1 | 13.11 | 14.35 | -14.35 | No |
| 2 | 14.44 | 11.94 | -11.94 | Yes |

calculate absolutes, and sort these values. The result presented in Table 11. We set I to 0 and compute the variance of the p most significant absolute contracts (using p = N/2, where N is the run size). The outcomes were presented in Table 12. The authors noticed that $\sigma_3^2$ is greater than $\sigma_2^2$, so we stopped and looked for the active factors. Stage 1's final values are $uml_i = 34.76$, $lml_i = -34.76$ and $\sigma_2^2 = 4$. Consequently, the following active factors exist outside the critical region: $h_{18}, h_{12}, h_5, h_3, h_4, h_{15}, h_9, h_{10}$ and $h_{11}$.

## 3.3. Analysis results of the edges design method

In this section, a ready-made edge design is selected from a published scientific paper consisting of six factors and 12 runs (N) [14,15]. This design was examined horizontally to ensure agreement with the questionnaire's design. The chosen design was then analyzed by designing the above edges. The design chosen from the scientific literature is as follows.

**Table 7. The third supersaturated application of the questionnaire.**

| $h_1$ | + | + | + | + | + | + | + | + | + | + | + | + | + | + | + | + | + | + |
|---|---|---|---|---|---|---|---|---|---|---|---|---|---|---|---|---|---|---|
| $h_2$ | + | + | - | + | + | + | + | + | + | + | + | - | + | - | - | + | + |
| $h_3$ | + | - | - | - | + | - | - | + | + | - | + | - | + | - | + | + | - |
| $h_4$ | + | - | - | - | + | - | - | - | + | - | + | - | + | - | + | + | - |
| $h_5$ | - | + | - | - | - | + | - | - | + | - | - | - | + | - | - | - | - |
| $h_6$ | - | + | - | + | + | - | + | + | + | + | - | + | + | + | + | - | - |
| $h_7$ | - | + | - | + | + | - | - | + | + | - | - | + | + | + | - | - | - |
| $h_8$ | + | + | + | + | + | - | + | + | + | - | + | + | + | + | + | - | - |
| $h_9$ | + | + | + | + | + | + | + | + | + | + | + | + | + | - | - | - | + |
| $h_{10}$ | + | + | - | + | + | + | - | + | + | - | + | - | + | + | + | + | + |
| $h_{11}$ | + | + | - | + | + | - | + | + | + | - | + | + | + | + | + | + | + |
| $h_{12}$ | - | + | - | - | - | - | + | + | + | - | + | - | + | - | + | + | - |
| $h_{13}$ | - | + | - | + | + | + | + | + | + | + | + | + | + | + | + | + | + |
| $h_{14}$ | - | + | - | - | - | + | + | + | + | + | + | - | + | - | + | - | + |
| $h_{15}$ | - | + | - | + | - | - | + | + | + | - | + | - | + | - | - | - | - |
| $h_{16}$ | - | + | + | - | + | - | + | - | + | - | - | - | + | + | + | - | - |
| $h_{17}$ | + | + | + | + | + | - | + | + | + | + | + | + | + | + | + | - | + |
| $h_{18}$ | - | + | - | - | - | - | - | - | + | - | + | - | + | - | - | - | - |
| Y | 2 | 3 | 0 | 2 | 1 | 0 | 3 | 1 | 18 | 2 | 2 | 1 | 3 | 4 | 1 | 1 | 1 |

**Table 8. Sorted absolute contrasts for Application 3.**

| J | 1 | 2 | 3 | 4 | 5 | 6 | 7 | 8 | 9 | 10 | 11 | 12 | 13 | 14 | 15 | 16 | 17 | 18 |
|---|---|---|---|---|---|---|---|---|---|---|---|---|---|---|---|---|---|---|
| $h_n$ | $h_1$ | $h_{17}$ | $h_9$ | $h_{11}$ | $h_{13}$ | $h_8$ | $h_2$ | $h_6$ | $h_{10}$ | $h_{14}$ | $h_7$ | $h_{16}$ | $h_{12}$ | $h_{15}$ | $h_3$ | $h_4$ | $h_5$ | $h_{18}$ |
| $\|M(j)\|$ | 45 | 43 | 41 | 41 | 41 | 37 | 33 | 33 | 33 | 23 | 21 | 21 | 19 | 19 | 13 | 11 | 7 | 3 |

**Table 9. Step-by-step calculations for Application 3.**

| i | $\sigma_i^2$ | $uml_i$ | $lml_i$ | $\sigma_i^2 > \sigma_{i-1}^2$ |
|---|---|---|---|---|
| 0 | 21.77 | **35.19** | **-35.19** | - |
| 1 | **40.11** | **31.63** | **-31.63** | Yes |
| 2 | **14.44** | **11.94** | **-11.94** | Yes |

**3.3.1. The first design with the edges of the questionnaire.** The following is an edge design analysis of the data in Table 13. To begin, Table 14 shows that all six contrasts of response y over the edges and the absolute regard are present. Second, we computed the center to forecast the number p as a powerful part. Third, we discovered ($\sigma$), w (p), and k^2^0.5. Finally, if w (p) for some hypothesis p is more critical than p, the method is terminated, and a unique factor is sought. The results are shown in Table 15; We have w (2) = 1, indicating a unique factor, which is $h_8$ (Recurrent urinary tract infection).

**3.3.2. The second design with the edges of the questionnaire.** The following is an edge design analysis of the data in Table 16. All six contrasts of response y over the edges and the absolute regard are presented in Table 17. Second, we computed the center to forecast the number p as powerful parts. Third, we discovered ($\sigma$), w (p), and k^2^0.5. Finally, if the w (p) for some hypothesis p is more critical than p, the method is terminated, and a unique factor is sought. The results are listed in Table 18; where w (5) = 4., indicating that there are unique

**Table 10. The fourth supersaturated application of the questionnaire.**

| $h_1$ | + | - | + | + | + | - | + | + | + | + | - | + | + | + | + | + | + |
|---|---|---|---|---|---|---|---|---|---|---|---|---|---|---|---|---|---|
| $h_2$ | + | - | + | + | + | - | + | - | + | + | - | + | + | + | + | - | + |
| $h_3$ | - | + | + | - | - | - | + | + | + | + | - | - | - | - | + | - | - |
| $h_4$ | - | + | + | - | - | - | + | + | - | + | - | - | + | - | + | - | - |
| $h_5$ | - | + | - | - | - | - | + | + | - | + | - | - | - | - | - | - | - |
| $h_6$ | - | + | - | + | + | - | + | + | + | + | - | + | + | - | + | + | + |
| $h_7$ | - | - | - | - | + | - | + | + | + | + | - | + | + | - | + | + | + |
| $h_8$ | + | - | + | - | - | - | + | + | + | + | - | + | + | - | + | + | + |
| $h_9$ | + | + | + | + | + | - | + | + | + | + | - | + | + | + | + | + | + |
| $h_{10}$ | + | + | + | + | + | - | + | + | + | + | - | + | - | + | + | + | + |
| $h_{11}$ | + | + | + | - | + | - | + | + | + | + | - | + | + | + | + | + | + |
| $h_{12}$ | - | - | - | + | - | - | + | + | + | + | - | + | + | - | - | - | - |
| $h_{13}$ | + | - | + | + | + | - | + | + | + | + | - | + | + | + | + | + | + |
| $h_{14}$ | + | - | - | + | + | - | + | + | + | + | - | + | + | + | + | + | + |
| $h_{15}$ | - | + | + | - | - | - | + | + | + | + | - | + | + | - | + | - | - |
| $h_{16}$ | - | - | - | - | + | - | + | - | - | + | - | + | + | - | - | + | - |
| $h_{17}$ | - | + | + | - | + | - | + | + | + | + | - | + | + | + | + | + | - |
| $h_{18}$ | - | + | - | - | - | - | + | - | - | + | - | - | - | - | + | - | - |
| Y | 25 | 1 | 1 | 0 | 9 | 2 | 1 | 3 | 0 | 1 | 6 | 0 | 0 | 1 | 0 | 0 | 1 |

**Table 11. Sorted absolute contrasts for Application 4.**

| J | 1 | 2 | 3 | 4 | 5 | 6 | 7 | 8 | 9 | 10 | 11 | 12 | 13 | 14 | 15 | 16 | 17 | 18 |
|---|---|---|---|---|---|---|---|---|---|---|---|---|---|---|---|---|---|---|
| $h_n$ | $h_{18}$ | $h_{12}$ | $h_5$ | $h_3$ | $h_4$ | $h_{15}$ | $h_9$ | $h_{10}$ | $h_{11}$ | $h_1$ | $h_{13}$ | $h_{14}$ | $h_{16}$ | $h_2$ | $h_7$ | $h_6$ | $h_{17}$ | $h_8$ |
| $|M(j)|$ | 45 | 41 | 39 | 37 | 37 | 37 | 35 | 35 | 35 | 33 | 33 | 31 | 29 | 27 | 21 | 19 | 17 | 13 |

**Table 12. Step-by-step calculations for Application 4.**

| i | $\sigma_i^2$ | $uml_i$ | $lml_i$ | $\sigma_i^2 > \sigma_{i-1}^2$ |
|---|---|---|---|---|
| 0 | 11.11 | 37.99 | -37.99 | - |
| 1 | 5.77 | 35.92 | -35.92 | No |
| 2 | 4 | 34.76 | -34.76 | No |
| 3 | 4.44 | 32.50 | -32.50 | Yes |

| | | | | | |
|---|---|---|---|---|---|
| 1 | 1 | -1 | 1 | 1 | 1 |
| -1 | 1 | 1 | 1 | 1 | 1 |
| 1 | -1 | 1 | 1 | 1 | 1 |
| -1 | -1 | -1 | 1 | -1 | 1 |
| -1 | -1 | -1 | 1 | 1 | -1 |
| -1 | -1 | -1 | -1 | 1 | 1 |
| -1 | 1 | -1 | 1 | 1 | 1 |
| -1 | -1 | 1 | 1 | 1 | 1 |
| 1 | -1 | -1 | 1 | 1 | 1 |
| -1 | -1 | -1 | -1 | -1 | 1 |
| -1 | -1 | -1 | 1 | -1 | -1 |
| -1 | -1 | -1 | -1 | 1 | -1 |

**Table 13. The first design with the edges of the questionnaire.**

| N\ Factor | $h_3$ | $h_4$ | $h_5$ | $h_6$ | $h_7$ | $h_8$ | Y |
|---|---|---|---|---|---|---|---|
| 1 | 1 | 1 | -1 | 1 | 1 | 1 | 1 |
| 2 | -1 | 1 | 1 | 1 | 1 | 1 | 2 |
| 3 | 1 | -1 | 1 | 1 | 1 | 1 | 2 |
| 4 | -1 | -1 | -1 | 1 | -1 | 1 | 2 |
| 5 | -1 | -1 | -1 | 1 | 1 | -1 | 2 |
| 6 | -1 | -1 | -1 | -1 | 1 | 1 | 2 |
| 7 | -1 | 1 | -1 | 1 | 1 | 1 | 1 |
| 8 | -1 | -1 | 1 | 1 | 1 | 1 | 3 |
| 9 | 1 | -1 | -1 | 1 | 1 | 1 | 1 |
| 10 | -1 | -1 | -1 | -1 | -1 | 1 | 1 |
| 11 | -1 | -1 | -1 | 1 | -1 | -1 | 3 |
| 12 | -1 | -1 | -1 | -1 | 1 | -1 | 0 |

**Table 14. Model-free tests with an edge plan.**

| $h_3$ | $h_4$ | $h_5$ | $h_6$ | $h_7$ | $h_8$ |
|---|---|---|---|---|---|
| 0 | -1 | 1 | 1 | -1 | 2 |
| 0 | 1 | 1 | 1 | 1 | 2 |

**Table 15. Step estimations for the edge plan investigation.**

| $p$ | Median | $\sigma(p)$e | $k \times 2^{0.5}\sigma(p)$ | $\omega(p)$ | $\omega(p) < p$? |
|---|---|---|---|---|---|
| 0 | 1 | 1.047565602 | 1.481481481 | 1 | No |
| 1 | 1 | 1.047565602 | 1.481481481 | 1 | No |
| 2 | 1 | 1.047565602 | 1.481481481 | 1 | Yes |

factors $h_6$ (Lack of exercise), $h_7$ (Obesity), $h_8$ (Recurrent urinary tract infection) and $h_{11}$ (presence of kidney stones).

## 4. Discussion and conclusion

CKD is a global health issue that can lead to severe complications such as kidney failure and death. It affects 195 million women worldwide each year and is currently the eighth leading cause of death in women, accounting for 600,000 deaths annually. This study aimed to identify the actual causes that leading to an increase in kidney failure cases using supersaturated and edge design analysis methods. Models of designs were used for each method using applications resulting from the questionnaire to achieve this goal. Applications were used for each technique in the analysis. The analysis method using supersaturated designs revealed that the reasons for the increase in kidney failure cases were as follows:

$h_9, h_8, h_1, h_6, h_{14}, h_{18}, h_{10}, h_{13}, h_2, h_{17}, h_{11}, h_{12}, h_5, h_3, h_4, h_{15}$ and $h_{11}$. At the same time, the design analysis method by edges gave that the reasons that led to an increase in kidney failure cases: $h_8, h_6, h_7$, and $h_{11}$. Finally, the similar reasons in the two methods are $h_6$ (Lack of exercise), $h_8$ (Recurrent urinary tract infection), and $h_{11}$ (The presence of kidney stones). The Saudi government, represented by the Ministry of Health, should publish periodic and cultural publications to educate the community about these reasons, urging them to practice sports and the early detection of stones and urinary tract infections.

**Table 16. The second design with the edges of the questionnaire.**

| N\ Factor | $h_6$ | $h_7$ | $h_8$ | $h_9$ | $h_{10}$ | $h_{11}$ | Y |
|---|---|---|---|---|---|---|---|
| 1 | 1 | 1 | -1 | 1 | 1 | 1 | 2 |
| 2 | -1 | 1 | 1 | 1 | 1 | 1 | 2 |
| 3 | 1 | -1 | 1 | 1 | 1 | 1 | 2 |
| 4 | -1 | -1 | -1 | 1 | -1 | 1 | 2 |
| 5 | -1 | -1 | -1 | 1 | 1 | -1 | 0 |
| 6 | -1 | -1 | -1 | -1 | 1 | 1 | 4 |
| 7 | -1 | 1 | -1 | 1 | 1 | 1 | 1 |
| 8 | -1 | -1 | 1 | 1 | 1 | 1 | 1 |
| 9 | 1 | -1 | -1 | 1 | 1 | 1 | 3 |
| 10 | -1 | -1 | -1 | -1 | -1 | 1 | 2 |
| 11 | -1 | -1 | -1 | 1 | -1 | -1 | 0 |
| 12 | -1 | -1 | -1 | -1 | 1 | -1 | 0 |

**Table 17. Model-free tests with an edge plan.**

| $h_6$ | $h_7$ | $h_8$ | $h_9$ | $h_{10}$ | $h_{11}$ |
|---|---|---|---|---|---|
| 1 | 1 | -1 | 0 | 0 | 4 |
| 1 | 1 | 1 | 0 | 0 | 4 |

**Table 18. Step estimations for the edge plan investigation.**

| $p$ | Median | $\sigma(p)$ e | $k \times 2^{0.5}\sigma(p)$ | $\omega(p)$ | $\omega(p) < p$? |
|---|---|---|---|---|---|
| 0 | 1 | 1.047565602 | 1.481481481 | 1 | No |
| 1 | 1 | 1.047565602 | 1.481481481 | 1 | No |
| 2 | 0.5 | 0.523782801 | 0.740740741 | 4 | No |
| 3 | 0 | 0 | 0 | 4 | No |
| 4 | 0 | 0 | 0 | 4 | No |
| 5 | 0 | 0 | 0 | 4 | Yes |

## Supporting information

**S1 Data.**
(PDF)

## Acknowledgments

This research was funded by the Scientific Research Deanship at the University of Ha'il, Saudi Arabia, project number RD-21 001.

## Author Contributions

**Data curation:** Alanazi Talal Abdulrahman.

**Formal analysis:** Alanazi Talal Abdulrahman.

**Investigation:** Alanazi Talal Abdulrahman, Dalia Kamal Alnagar.

**Methodology:** Alanazi Talal Abdulrahman.

**Resources:** Alanazi Talal Abdulrahman.

**Software:** Alanazi Talal Abdulrahman.

**Supervision:** Alanazi Talal Abdulrahman.

**Writing – original draft:** Alanazi Talal Abdulrahman.

**Writing – review & editing:** Alanazi Talal Abdulrahman, Dalia Kamal Alnagar.

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
