## [Decision Letter · Decision Letter 0]

21 Feb 2024

PONE-D-23-22130Accurate statistical methods to cover the aspects of the increase in the incidence of kidney failure: a survey study in Saudi ArabiaPLOS ONE

Dear Dr. Abdulrahman,

Thank you for submitting your manuscript to PLOS ONE. After careful consideration, we feel that it has merit but does not fully meet PLOS ONE’s publication criteria as it currently stands. Therefore, we invite you to submit a revised version of the manuscript that addresses the points raised during the review process.

We look forward to receiving your revised manuscript.

Kind regards,

V. Vinoth Kumar

Academic Editor

PLOS ONE

Journal Requirements:

   "There are no founding"

5. In the online submission form, you indicated that "The data supporting the study's findings are available upon request from the corresponding author"

7. Please ensure that you refer to Figure 4 in your text as, if accepted, production will need this reference to link the reader to the figure.

Reviewers' comments:

Reviewer's Responses to Questions

**Comments to the Author**

1. Is the manuscript technically sound, and do the data support the conclusions?

Reviewer #1: Partly

Reviewer #2: Partly

2. Has the statistical analysis been performed appropriately and rigorously? 

Reviewer #1: Yes

Reviewer #2: Yes

3. Have the authors made all data underlying the findings in their manuscript fully available?

Reviewer #1: Yes

Reviewer #2: Yes

4. Is the manuscript presented in an intelligible fashion and written in standard English?

Reviewer #1: Yes

Reviewer #2: Yes

5. Review Comments to the Author

Reviewer #1: 1. Related works must be presented elaboratively.

2. Number of references needs to be increased

3. Significance of the work needs to be highlighted clearly

4. English language needs to be rechecked and minor corrections are necessary.

Reviewer #2: 1. Abstract: result section: the ℎ9, ℎ8, ℎ1,

ℎ6, ℎ14, ℎ18, need to be clarified for the readers. I suggest the authors should use the actual findings so as to be easily identified by the readers.

2. body of the paper: result section: the repetition of the words [[EQUATION]], [[EQUATION]] may not be understood to the readers, can use other forms of findings, so as to be easily read by the readers? Also this is observed in he conclusion section.

3. The author should identify the validity of the two methods: Analysis by supersaturated designs and Analysis by design with edges. Are these methods compared to other developed methods?

6. PLOS authors have the option to publish the peer review history of their article (what does this mean?). If published, this will include your full peer review and any attached files.

Reviewer #1: **Yes: **Bharanidharan N

Reviewer #2: No

---

## [Author Response · Author response to Decision Letter 0]

25 Apr 2024

Dear Reviewer, 1

I would like to say thank you for the very excellent comments, which add a very wonderful addition to the scientific paper. The following table shows the addition by the authors.

Reviewer 1 comments Authors comments 

Related works must be presented elaboratively Done in third paragraph from introduction section

Number of references needs to be increased Done we added refences 6, 7, 8, 9 and 15

Significance of the work needs to be highlighted clearly Done in second paragraph from introduction section that: The study aims to identify the actual reasons…

English language needs to be rechecked and minor corrections are necessary Done. We highlighted as follow: 

- Line 3, 4, 5, 18 in abstract 

- Line 3, 5 , 27 in introduction 

- Subsection 2.3.1 step 5 : differences.

- Third line in conclusion.

Dear Reviewer, 2

I would like to say thank you for the very excellent comments, which add a very wonderful addition to the scientific paper. The following table shows the addition by the authors.

Reviewer 2 comments Authors comments 

Abstract: result section: the ℎ9, ℎ8, ℎ1,

ℎ6, ℎ14, ℎ18, need to be clarified for the readers. I suggest the authors should use the actual findings to be easily identified by the readers. Done. We use the actual finding in abstract 

body of the paper: result section: the repetition of the words [[EQUATION]], [[EQUATION]] may not be understood to the readers, can use other forms of findings, so as to be easily read by the readers? Also this is observed in the conclusion section. Done in subsection 2.3.1

The author should identify the validity of the two methods: Analysis by supersaturated designs and Analysis by design with edges. Are these methods compared to other developed methods? Done in fourth paragraph from introduction section that The validity of the two methods….

Authors 

Dr Talal 

Dr Dalia

---

## [Decision Letter · Decision Letter 1]

18 Jun 2024

PONE-D-23-22130R1Accurate statistical methods to cover the aspects of the increase in the incidence of kidney failure: a survey study in Ha'il -Saudi ArabiaPLOS ONE

Dear Dr. Abdulrahman,

Thank you for submitting your manuscript to PLOS ONE. After careful consideration, we feel that it has merit but does not fully meet PLOS ONE’s publication criteria as it currently stands. Therefore, we invite you to submit a revised version of the manuscript that addresses the points raised during the review process.

We look forward to receiving your revised manuscript.

Kind regards,

V. Vinoth Kumar

Academic Editor

PLOS ONE

Journal Requirements:

Reviewers' comments:

Reviewer's Responses to Questions

**Comments to the Author**

1. If the authors have adequately addressed your comments raised in a previous round of review and you feel that this manuscript is now acceptable for publication, you may indicate that here to bypass the “Comments to the Author” section, enter your conflict of interest statement in the “Confidential to Editor” section, and submit your "Accept" recommendation.

Reviewer #1: All comments have been addressed

Reviewer #2: All comments have been addressed

2. Is the manuscript technically sound, and do the data support the conclusions?

Reviewer #1: Yes

Reviewer #2: Partly

3. Has the statistical analysis been performed appropriately and rigorously? 

Reviewer #1: Yes

Reviewer #2: Yes

4. Have the authors made all data underlying the findings in their manuscript fully available?

Reviewer #1: Yes

Reviewer #2: Yes

5. Is the manuscript presented in an intelligible fashion and written in standard English?

Reviewer #1: Yes

Reviewer #2: Yes

6. Review Comments to the Author

Reviewer #1: All suggestions are incorporated by the authors.

Only minor language corrections are required before publishing the manuscript.

Reviewer #2: There minor comments in methodology sections. In section 2.4: Combine the results of the two methods. The authors use the future tense, I suggest to use the past tense since the procedure or tests had already been applied. Secondly in abstract section the sentence(All variables are studied from to) is incomplete.

7. PLOS authors have the option to publish the peer review history of their article (what does this mean?). If published, this will include your full peer review and any attached files.

Reviewer #1: No

Reviewer #2: **Yes: **Moawia Gameraddin

---

## [Author Response · Author response to Decision Letter 1]

18 Jul 2024

Dear Reviewers

I would like to say thank you for the very excellent comments, which add a very wonderful addition to the scientific paper. The following table shows the addition by the authors in the attachment

---

## [Decision Letter · Decision Letter 2]

8 Aug 2024

Accurate statistical methods to cover the aspects of the increase in the incidence of kidney failure: a survey study in Ha'il -Saudi Arabia

PONE-D-23-22130R2

Dear Dr. Abdulrahman,

We’re pleased to inform you that your manuscript has been judged scientifically suitable for publication and will be formally accepted for publication once it meets all outstanding technical requirements.

Kind regards,

V. Vinoth Kumar

Academic Editor

PLOS ONE

Additional Editor Comments (optional):

Reviewers' comments:

Reviewer's Responses to Questions

**Comments to the Author**

1. If the authors have adequately addressed your comments raised in a previous round of review and you feel that this manuscript is now acceptable for publication, you may indicate that here to bypass the “Comments to the Author” section, enter your conflict of interest statement in the “Confidential to Editor” section, and submit your "Accept" recommendation.

Reviewer #2: All comments have been addressed

2. Is the manuscript technically sound, and do the data support the conclusions?

Reviewer #2: Yes

3. Has the statistical analysis been performed appropriately and rigorously? 

Reviewer #2: Yes

4. Have the authors made all data underlying the findings in their manuscript fully available?

Reviewer #2: Yes

5. Is the manuscript presented in an intelligible fashion and written in standard English?

Reviewer #2: Yes

6. Review Comments to the Author

Reviewer #2: All the comments have been satisfied. In methodology section, I suggest the authors to validate the Methods which were used in the analysis.

7. PLOS authors have the option to publish the peer review history of their article (what does this mean?). If published, this will include your full peer review and any attached files.

Reviewer #2: No

---

## [Editor Report · Acceptance letter]

19 Aug 2024

PONE-D-23-22130R2 

PLOS ONE

Dear Dr. Abdulrahman, 

I'm pleased to inform you that your manuscript has been deemed suitable for publication in PLOS ONE. Congratulations! Your manuscript is now being handed over to our production team.

Kind regards, 

on behalf of

Dr. V. Vinoth Kumar 

Academic Editor

PLOS ONE